# Gbb Regulates Blood Cell Proliferation and Differentiation through JNK and EGFR Signaling Pathways in the *Drosophila* Lymph Gland

**DOI:** 10.3390/cells12040661

**Published:** 2023-02-19

**Authors:** Wenhao Zhang, Dongmei Wang, Jingjing Si, Lihua Jin, Yangguang Hao

**Affiliations:** 1Department of Genetics, College of Life Sciences, Northeast Forestry University, Harbin 150040, China; 2Department of Basic Medical, Shenyang Medical College, Shenyang 110034, China

**Keywords:** hematopoiesis, lymph gland, Gbb, JNK, EGFR

## Abstract

The *Drosophila* lymph gland is an ideal model for studying hematopoiesis, and unraveling the mechanisms of *Drosophila* hematopoiesis can improve our understanding of the pathogenesis of human hematopoietic malignancies. Bone morphogenetic protein (BMP) signaling is involved in a variety of biological processes and is highly conserved between *Drosophila* and mammals. Decapentaplegic (Dpp)/BMP signaling is known to limit posterior signaling center (PSC) cell proliferation by repressing the protooncogene *dmyc*. However, the role of two other TGF-β family ligands, Glass bottom boat (Gbb) and Screw (Scw), in *Drosophila* hematopoiesis is currently largely unknown. Here, we showed that the loss of Gbb in the cortical zone (CZ) induced lamellocyte differentiation by overactivation of the EGFR and JNK pathways and caused excessive differentiation of plasmatocytes, mainly by the hyperactivation of EGFR. Furthermore, we found that Gbb was also required for preventing the hyperproliferation of the lymph glands by inhibiting the overactivation of the Epidermal Growth Factor Receptor (EGFR) and c-Jun N-terminal Kinase (JNK) pathways. These results further advance our understanding of the roles of Gbb protein and the BMP signaling in *Drosophila* hematopoiesis and the regulatory relationship between the BMP, EGFR, and JNK pathways in the proliferation and differentiation of lymph gland hemocytes.

## 1. Introduction

Recent studies have revealed that many signaling pathways controlling hematopoiesis and innate immunity are highly conserved between mammals and *Drosophila*. The signaling pathways that regulate mammalian bone marrow hematopoiesis, such as the Dpp/BMP, JNK, Janus kinase/signal transducer and activator of transcription (JAK–STAT), and insulin signaling pathways, were also found to control the regeneration and differentiation of *Drosophila* lymph gland hematopoietic stem cells [1]. Moreover, some signaling pathways that cause leukemogenesis, such as the activation of JAK–STAT, can induce a leukemia-like phenotype in *Drosophila*, including the excessive growth of the lymph glands, an increased number of circulating hemocytes, and melanoma production [2,3]. In addition, the BCR–ABL, Tax-1, RUNX1, and NUP98–HOXA9 (NA9) fusion proteins associated with chronic granulocytic leukemia (CML) and acute myeloid or lymphoid leukemia (AML/ALL) have also been found to cause leukemia-like traits in *Drosophila* lymph glands [4,5,6,7]. As *Drosophila* is amenable to genetic manipulation and has a short growth cycle, it has increasingly become a genetic model for studying hematopoiesis, leukemia, and natural immunity [1].

Similar to that in vertebrates, hematopoiesis in *Drosophila* occurs in two stages of development. The first type of hematopoiesis is derived from the mesoderm of the embryonic head, and the second stage of hematopoiesis occurs in the lymph glands of the larvae [8,9]. The lymph gland is the hematopoietic organ of *Drosophila* and contains a pair of primary lobes and a series of posterior lobes. There are three distinct zones within the anterior lobe: the cortical zone (CZ), the medullary zone (MZ), and the posterior signaling center (PSC). The mature plasmatocytes and crystal cells are located in the CZ and can be defined by the expression of Nimrod1 (NimC1/P1) and Hindsight (Hnt), respectively [10]. The prohemocytes (stem cell-like precursor blood cells) within the MZ can be defined by markers such as Domeless (Dome), Patched (Ptc), and DE-cadherin (DE-cad), whose differentiation fate depends on intrinsic factors and extensive intercellular interactions in the microenvironment [10,11]. The PSC region serves as a hematopoietic niche to maintain the normal differentiation of prohemocytes [12]. The lamellocytes are also a type of mature blood cells; they are not present in normal lymph glands and are only induced by an immune response to an infection [13]. Moreover, the autonomic activation of immune signals, such as the JAK/STAT, JNK, Toll, and Ras/EGFR signaling pathways, can also induce the generation of lamellocytes [14,15,16,17].

Bone morphogenetic proteins (BMPs) are a functionally well-conserved class of growth factors belonging to the TGF-β superfamily. In *Drosophila,* there are three BMP ligands: Decapentaplegic (Dpp), which is the ortholog of vertebrate TGF-β superfamily ligands BMP 2/4; Screw (Scw), which appears to be a distantly related BMP with similarity to the activins; and Glass bottom boat (Gbb), which is the ortholog of the mammalian BMP 5/6/7/8 [18]. Dpp can bind to the BMP type I receptor Thickveins (Tkv), or Saxophone (Sax) and the type II receptor Punt, and it is involved in a variety of biological processes, including oogenesis, the patterning of the embryonic mesoderm, morphogenesis of the midgut, the imaginal disc and ventral head, and stem cell maintenance [19,20,21,22,23,24,25,26,27]. Of note, Dpp signaling also regulates hematopoiesis of the lymph gland via the limitation of PSC cell proliferation by antagonizing the activity of wingless (Wg)/Wnt signaling [28]. Similar to Dpp, Gbb has also been shown to regulate wing disc development, midgut formation, neuromuscular junctions, stem cell maintenance, and fat body metabolism [27,29,30,31,32,33]. However, Gbb is not required for PSC niche size control; the number of PSC cells was not changed in a *gbb* mutant [28]. Apart from this study, no other experiments have investigated the role of Gbb in the hematopoiesis of *Drosophila*.

In this study, we knocked down *gbb* in the hematopoietic system and determined the role of Gbb in the CZ. Our results showed that the loss of Gbb in CZ induced lamellocyte differentiation by the overactivation of the EGFR and JNK pathways and caused the excessive differentiation of plasmatocytes, mainly by the hyperactivation of the EGFR signaling pathway. Moreover, we found that Gbb was also required for preventing the hyperproliferation of the lymph glands by inhibiting the overactivation of the EGFR and JNK pathways. These results further advance our understanding of the roles of the Gbb protein and BMP signaling in hematopoiesis.

## 2. Materials and Methods

### 2.1. Fly Stocks and Culture

We used the following lines in this study: *gbb RNAi#1*(*THU201501092*), *gbb RNAi#2* (*THU1480*), *EGFR-DN,* and *Hml-Gal4*, which were obtained from the Tsinghua Fly Center; *Hml-Gal4*; *UAS-2xEGFP*, which was a gift from Utpal Banerjee; *UAS-puc*, which was a gift from José Carlos Pastor-Pareja; *gbb-GFP* (BDSC:63056), *w*^1118^, *Cg-Gal4*, and *Da-Gal4*, which were obtained from the Bloomington Drosophila Stock Center (BDSC); *Pxn-Gal4*; *UAS-GFP*, which was a gift from Norbert Perrimon; *dome-Gal4*; and *UAS-2xEGFP*, which was a gift from Jiwon Shim. The crosses involving RNAi lines were reared at 29 °C, and the other strains and crosses were reared at 25 °C. All strains and crosses were cultured on standard cornmeal–yeast media.

### 2.2. Immunostaining

For antibody staining, the lymph glands and circulating hemocytes of third-instar larvae were fixed and stained as described previously [11,34]. Briefly, 5–6 third-instar larvae (96 h after egg laying) were opened via an incision at both the posterior and anterior ends in 20 μL of phosphate-buffered saline (PBS), and the circulating hemocytes were bled and allowed to attach to a glass slide for 30 min. The following primary antibodies were used: rabbit anti-GFP (1:100, Thermo Fisher Scientific, Waltham, MA, USA), mouse anti-L1 (1:50), and mouse anti-P1 (1:50), which were gifts from I. Ando; rabbit anti-p-Erk (1:50, Cell Signaling Technology, Danvers, MA, USA); and rabbit anti-p-JNK (1:200, Promega, Madison, WI, USA). Mouse anti-dorsal (1:50, Developmental Studies Hybridoma Bank, Iowa City, IA, USA); rat anti-Stat92E (made in our lab); rabbit anti-PPO1 (1:100, a gift from Erjun Ling); rabbit anti-PH3 (1:200, Millipore, Burlington, MA, USA); Alexa Fluor 488-, Alexa Fluor 568-, and Alexa Fluor 594-conjugated secondary antibodies (Thermo Fisher Scientific, USA) and Hoechst (1:500, Sigma-Aldrich, Burlington, MA, USA) were used. All staining was performed in at least three independent experiments.

### 2.3. Image Analysis and Quantification

All images used for quantification were captured with a Zeiss Axioplan 2 microscope, and all quantification analyses were performed as described previously [11,34]. The total intensity value of p-Erk and p-JNK in each ROI (region of interest) with an identical threshold was captured and measured with ImageJ 1.47v. The ROIs in the fluorescent images were captured using the freehand tool and then converted to 8-bit images. For the quantification of the area of GFP+ and P1^+^ cells, the images were converted to eight bits and adjusted to obtain an identical threshold using ImageJ 1.47v. The area with an identical threshold was measured as the fluorescence+ area. For the quantification of the crystal cells index, the number of PPO1^+^ cells per relative unit area (the total number of PPO1^+^ cells/anterior lobe area) was calculated. For the quantification of the numbers of PH3^+^ cells, the total number of PH3^+^ cells in each primary lobe was counted with ImageJ. The third-instar larvae (96 h after egg laying) were used in all image analysis and quantification except for the quantification of PH3^+^ cells. For each genotype in each independent experiment, at least 10 lymph glands or at least 10 images of hemocytes were analyzed.

### 2.4. Quantitative Real-Time PCR

Total RNA from third-instar larvae was isolated with TRIzol (Invitrogen, Waltham, MA, USA). The obtained total RNA was used to generate cDNA with M-MLV Reverse Transcriptase RNase H Minus Point Mutant (Promega, Madison, WI, USA). Real-time PCR amplification was performed using a LightCycler 480 real-time PCR system (Roche, NY, USA) with FastStart Universal SYBR Green Master Mix (ROX) (Roche, USA). The following primers were used: *gbb*: F-GAGTGGCTGGTCAAGTCGAA and R-GAAGCCGATCATGAAGGGCT; *rp49*: F-AGTCGGATCGATATGCTAAGCTGT and R-TAACCGATGTTGGGCATCAGATACT. The results were normalized to the level of *rp49* mRNA in each sample. Three experiments per genotype were averaged.

### 2.5. Statistical Analysis

For the statistical analyses, the *p*-values were calculated with two-tailed unpaired Student’s *t*-tests or one-way ANOVAs using GraphPad Prism 6.0 software. The thresholds for statistical significance were established as * *p* < 0.05, ** *p* < 0.01, and *** *p* < 0.001, and a *p*-value > 0.05 indicated a nonsignificant difference. The data in all bar charts are shown as the means ± SD (error bar).

## 3. Results

### 3.1. Gbb Is Widely Expressed in the Drosophila Lymph Gland

To investigate the role of Gbb in the *Drosophila* lymph gland, we first identified the expression of the *gbb* gene and Gbb protein in the lymph gland. Bumsik et al.’s previously processed datasets of single-cell RNA-seq (http://big.hanyang.ac.kr/flyscrna (accessed on 25 July 2020)) were used to search the expression pattern of the *gbb* gene at the transcriptional level in the whole lymph gland [35], and we found that the *gbb* gene was widely expressed in most cell types of the lymph gland (Figure 1A,B). The endogenous localization of Gbb was surveyed using *Gbb-GFP*. Similar to the expression at the transcriptional level, we found that the Gbb protein was also expressed in the entire lymph gland, especially the posterior lobes, and was mainly located in the cytoplasm of hemocytes (Figure 1C,C′).

### 3.2. Knockdown of Gbb in the CZ Can Induce Lamellocyte Differentiation by Activating the EGFR and JNK Pathways

According to the expression pattern of Gbb in the lymph gland shown in the above results, Gbb is dramatically expressed in plasmatocytes, which are the major hemocyte type in the CZ. Thus, we first investigated the role of Gbb in the CZ. We knocked down *gbb* using the differentiated hemocyte-specific driver *Hml-Gal4* in the CZ and evaluated the changes in hemocyte differentiation. The anti-L1 antibody was used to examine the lamellocytes. We found that large numbers of lamellocytes appeared in the circulating hemolymph and lymph glands of *Hml > gbb RNAi#1* and *Hml > gbb RNAi#2* larvae (Figure 2A,B,E,F and Appendix A). Moreover, we used another differentiated hemocyte-specific driver, *Cg-Gal4*, to knock down *gbb* and found lamellocytes in the circulating hemolymph (Appendix A). To further determine whether the lamellocyte differentiation shown in the larvae with *gbb* gene knockdown depends on the expression levels of *gbb*, the transcription level of the *gbb* gene in *gbb* knockdown larvae was quantified. We used the ubiquitous driver *da-Gal4* to knock down *gbb* and found that the transcription level of *gbb* was reduced by nearly tenfold in the *gbb* knockdown larvae (Appendix A). These results confirmed that the loss of *gbb* can induce the generation of lamellocytes. Next, we further investigated the mechanism underlying lamellocyte differentiation after *gbb* knockdown. It is known that the excessive activation of some classical signaling pathways, such as Toll, JAK–STAT, JNK, and Ras/EGFR, can induce the generation of lamellocytes. Thus, we detected the activation of these pathways in the lymph glands and circulating hemolymph, respectively. The Toll signaling transcription factor Dorsal and the JAK–STAT transcription factor Stat92E were detected via antibody staining. We found that the localization and expression of Dorsal and Stat92E in *Hml > gbb RNAi#1* were not changed compared with those found in the control (Appendix A). Then, we focused on the Ras/EGFR and JNK pathways and stained lymph glands and circulating hemocytes with the anti-p-Erk antibody and anti-p-JNK antibody to detect the expression of the target of EGFR, p-Erk, and the target of the JNK pathway, p-JNK, respectively. Notably, both the p-Erk and p-JNK signals were significantly increased in the circulating hemocytes and lymph glands of *Hml > gbb RNAi#1* (Figure 2I–T). To further determine whether the generation of lamellocytes in *Hml > gbb RNAi#1* is caused by the overactivation of the EGFR and JNK pathways, the *Hml > gbb RNAi#1* line was crossed with *UAS-puc*, which is the negative regulator of the JNK pathway, and *UAS-EGFR-DN,* which expresses a dominant-negative EGFR. We found that the inactivation of the EGFR or JNK pathway effectively inhibited the formation of lamellocytes in the circulating hemocytes and lymph glands of *Hml > gbb RNAi#1* (Figure 2C,D,G,H). These results indicate that Gbb in the CZ restricts lamellocyte differentiation by preventing the hyperactivation of the EGFR and JNK pathways.

### 3.3. The Loss of Gbb in the CZ Can Induce Plasmatocyte Differentiation Primarily by Activating the EGFR Pathway but Not the JNK Pathway

Next, we examined the changes in the CZ upon the knockdown of Gbb using *Hml > UAS-GFP,* and the GFP-positive area represented the area of the CZ. The size of the GFP-positive area was larger in the lymph glands of *Hml > gbb RNAi#1* than those in the control (Figure 3A,B,I). Subsequently, we used the antibody against the mature plasmatocyte marker P1 to detect plasmatocyte differentiation. Similar to the Hml-GFP-positive area, the expansion of P1-positive plasmatocytes was also shown in *Hml > gbb RNAi#1* (Figure 3E,F,J). Moreover, we used another differentiated hemocyte-specific driver, *Pxn-Gal4,* to knock down *gbb* and found a similar expansion of Pxn-GFP-positive and P1-positive cells (Appendix A). These results suggest that Gbb in the CZ cell-autonomously controls plasmatocyte differentiation. Next, we further investigated the regulatory mechanism of Gbb in plasmatocyte differentiation. The above results show that the knockdown of Gbb in the CZ can induce lamellocyte differentiation by activating the EGFR and JNK pathways. We therefore first asked whether Gbb also controls the differentiation of plasmatocytes via both signaling pathways. As expected, the inactivation of the Ras/EGFR pathway can effectively inhibit the massive differentiation of plasmatocytes in *Hml > gbb RNAi#1* (Figure 3C,G,I,J). However, inhibiting the activation of the JNK pathways did not rescue the expansion of the Hml-GFP-positive area and only slightly prevented the expansion of the P1-positive area in *Hml > gbb RNAi#1* (Figure 3D,H,I,J). Taken together, these results indicate that Gbb in the CZ cell-autonomously prevents the excessive differentiation of plasmatocytes primarily by inhibiting the hyperactivation of the EGFR pathway but not the JNK pathway.

### 3.4. Knockdown of Gbb in Intermediate Progenitors and Progenitors Resulted in the Over-Differentiation of Crystal Cells and Plasmatocytes

We further detected the differentiation of crystal cells with the anti-PPO1 antibody. However, unlike the differentiation of plasmatocytes, the differentiation index of the crystal cells in *Hml > gbb RNAi* lymph glands was significantly reduced compared with that in the control (Figure 4A–C,G). Then, we used *Pxn-Gal4* to knock down *gbb* and found that the number of crystal cells in *Pxn > gbb RNAi* lymph glands was significantly increased compared with that in the control (Figure 4D–F,H). These results indicate that the expression pattern of *Hml-Gal4* and *Pxn-Gal4* may not be exactly the same, although both of them are maturation markers of hemocytes. In addition to being expressed in mature hemocytes, Pxn is also expressed in intermediate progenitors, initially described as being in a “transition state”, as these cells are both Dome+ and Pxn+ [36,37]. We next used the MZ progenitor-specific driver *dome-Gal4* to knock down *gbb,* and found that the loss of *gbb* in the MZ resulted in the massive differentiation of crystal cells and plasmatocytes; especially, the crystal cells were also observed in the posterior lobes of *dome > gbb RNAi#1* lymph glands (Figure 5A–H). Taken together, these results suggest that the knockdown of *gbb* only in the CZ maturing hemocytes can prevent the differentiation of crystal cells, but the knockdown of *gbb* in intermediate progenitors and progenitors results in the over-differentiation of crystal cells and plasmatocytes.

### 3.5. Gbb in the CZ Is Required for Preventing the Hyperproliferation of Lymph Glands by Inhibiting the Overactivation of EGFR and JNK Pathways

Of note, we observed that the loss of Gbb in the CZ resulted in a dramatic enlargement of the anterior lobes of the lymph glands (Figure 6A–C,J), suggesting that Gbb may regulate cell proliferation in the lymph gland. To confirm our speculation, the anti-PH3 antibody was used to examine the mitotic activity of the lymph glands at different larval stages. The increased PH3-positive cells were observed in the *Hml > gbb RNAi#1* and *Hml > gbb RNAi#2* lymph glands at 72 h and 96 h after egg laying (Figure 6D–I,K). The overactivation of the EGFR and JNK signaling pathways can promote excessive cell proliferation and tumorigenicity [16,38]. Thus, we asked whether Gbb regulates cell proliferation by inhibiting the activation of the EGFR and JNK pathways. Consistent with this hypothesis, the inactivation of the EGFR or JNK pathway dramatically reduced the overgrowth of the anterior lobes in the *Hml > gbb RNAi#1* lymph glands (Figure 7A–E). These results demonstrate that Gbb expression in the CZ is required for preventing the hyperproliferation of the lymph glands by inhibiting the overactivation of the EGFR and JNK pathways.

## 4. Discussion

The BMP signaling pathway is highly conserved between *Drosophila* and mammals, and *Drosophila* has become a valuable system to study BMPs due to the high functional conservation of the pathway and the molecular genetic tools available. It has been shown that BMP signaling is well known for its role in controlling proliferation in imaginal tissues and maintaining germline stem cells in the ovaries [21,39,40,41,42]. Recent studies have indicated that BMP signaling modulates the *Drosophila* immune response following bacterial infection, wounding, and parasitic nematodes [43,44,45]. NF-κB transcription factors are required for the activation of the BMP signaling pathway in nematode-infected flies [46]. Gbb and its receptor Wishful Thinking (BMPRII) are necessary for injury-induced allodynia in *Drosophila* [47]. Furthermore, injury can stimulate the production of Dpp and Gbb, which drive an expansion of intestinal stem cells (ISCs) by promoting their symmetric self-renewing division in the adult *Drosophila* midgut [48]. The TGF-β signaling pathway is involved in inflammation and tissue repair in mammals, and the lack of TGF-β signaling can affect the function, proliferation, and differentiation of immune cells [49,50]. Moreover, previous studies have suggested that suppressing BMP receptor 1A in mouse bone marrow stroma can cause an increased osteoblast count [39], and BMP4 was shown to be expressed in and regulate the mouse HSC [51]. Similarly, the BMP signaling pathway also controls the size of the *Drosophila* hematopoietic niche PSC. Dpp antagonizes the activity of wingless (Wg)/Wnt signaling, which positively regulates the number of PSC cells via the control of Dmyc expression [28]. Furthermore, this study also suggests that Gbb is not required for the size control of the PSC of the lymph gland.

Here, we further determined the role of Gbb in *Drosophila* hematopoiesis. We showed that Gbb is mainly expressed in the cytoplasm of the lymph gland hemocytes and cell-autonomously regulates the differentiation of the lymph glands in the CZ via multiple regulatory mechanisms. Consistent with its subcellular localization in the lymph glands, the cytoplasmic localization of Gbb has been found in neurons and intestinal cells in previous studies [47,52]. We found that the knockdown of *gbb* in the CZ significantly induced the abnormal differentiation of lamellocytes and plasmatocytes. However, the crystal cell number was significantly reduced in the lymph glands of *Hml > gbb RNAi*, probably because the knockdown of *gbb* triggers lamellocyte differentiation at the expense of crystal cells. Similar to these results, our previous studies also have shown that the loss of *jumu* or ectopic expression of *col* favors lamellocyte differentiation at the expense of crystal cells [11]. Moreover, the knockdown of *gbb* in intermediate progenitors and progenitors also induced the over-differentiation of crystal cells and plasmatocytes. Consistent with these results, a recent study showed that knocking down *Dlp* in the progenitors, which regulate Dpp signaling by stabilizing Dpp at the cell membranes, also increased blood cell differentiation and decreased the progenitor pool [53]. The role of Dpp in the PSC and MZ has been evaluated [28,53]; however, the function of this ligand in CZ is still unknown. In *Drosophila*, two other characterized BMP family ligands, Gbb and Scw, can form heterodimers with Dpp to augment the level and increase the range of BMP signaling in different cells and tissues [54]. Whether Gbb regulates the differentiation of the lymph gland hemocytes by forming heterodimers with Dpp and the function of Dpp in the CZ remain to be addressed.

Previous studies have shown that the JNK and EGFR signaling pathways participate in lamellocyte formation [16,55]. Our recent study further elucidated the role of the JNK and Ras/EGFR pathways in the CZ of the lymph glands [17]. Using *UAS-hep^Act^* to activate the JNK signaling pathway in the CZ can induce the generation of lamellocytes. However, the overexpression of *bsk* or Ras^v12^ (an activated form of Ras) in the CZ failed to induce lamellocyte formation, but overexpressing both genes simultaneously could lead to the production of lamellocytes [17]. These results indicated that the Ras/EGFR pathway in the CZ can cooperate with the JNK pathway to regulate the differentiation of lamellocytes. Here, we found that the loss of Gbb in the CZ promoted the phosphorylation of Erk and JNK, and inhibiting EGFR or JNK activation can effectively rescue the lamellocyte formation caused by the loss of Gbb. These results suggest that the BMP pathway may control the differentiation of lamellocytes by preventing the hyperactivation of EGFR and JNK signaling. Moreover, our study also showed that Gbb expression in the CZ is required for preventing the hyperproliferation of the lymph glands by inhibiting the overactivation of the EGFR and JNK pathways. It has been demonstrated that EGFR and JNK signaling can promote growth and proliferation in many cell types, and genetic hyperactivation of both signaling pathways can drive tumor formation [56,57,58,59]. Consistent with our results, a recent study showed that the loss of BMP induced tumorigenesis and consequently led to the aberrant activation of JNK/Mmp2 signaling, followed by intestinal barrier dysfunction and commensal imbalance [60]. Our present results provide a better understanding of the regulatory mechanism of the BMP signaling pathway in *Drosophila* hematopoiesis and important insights into the regulatory relationships of the BMP, EGFR, and JNK pathways in human hematopoietic malignancies. Furthermore, future studies will address the role of other components of the BMP pathway in *Drosophila* hematopoiesis.

## Figures and Tables

**Figure 1 cells-12-00661-f001:**
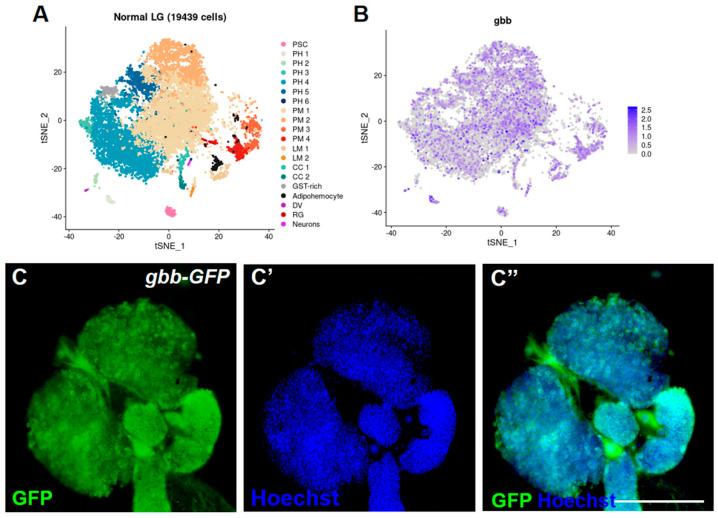
Gbb is widely expressed in the *Drosophila* lymph gland. (**A**) The t−SNE plot shows the two−dimensional projection of major cell type subclusters identified in the scRNA−seq of normal 120 h AEL (after egg laying) lymph glands. (**B**) Relative expression level of *gbb* selected in scRNA−seq of normal 120 h AEL lymph gland dataset corresponding to the t−SNE plot in A. (**C**–**C″**) Immunostaining against GFP shows that Gbb is expressed in the entire lymph gland. Scale bars: 100 μm (**C**–**C″**).

**Figure 2 cells-12-00661-f002:**
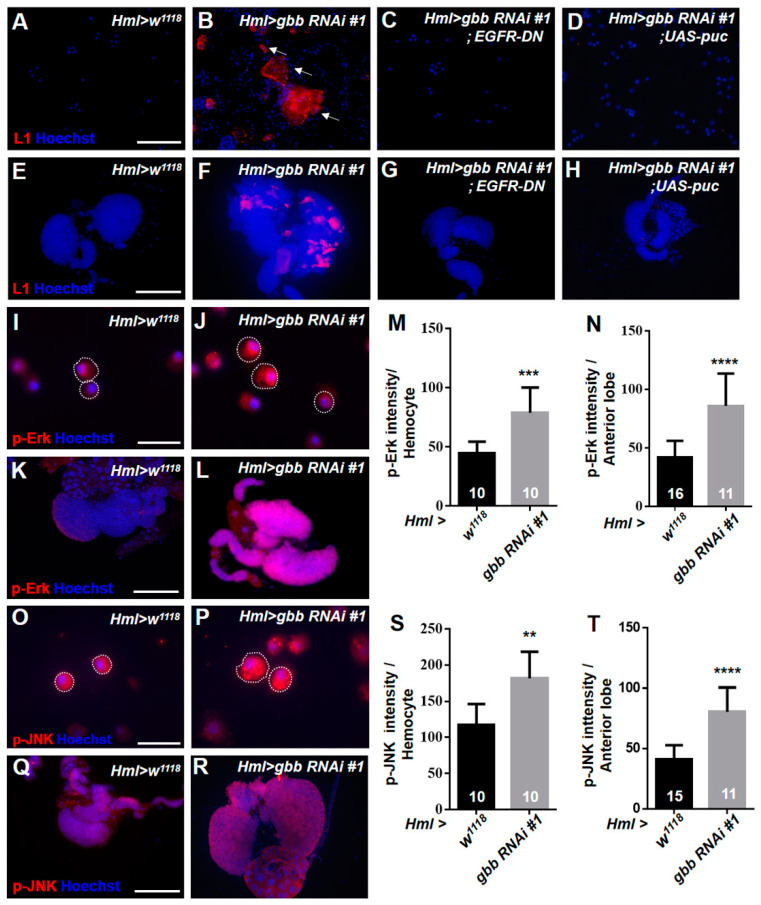
The knockdown of *gbb* in the CZ induces the differentiation of lamellocytes by activating the EGFR and JNK pathways. (**A**–**H**) Immunostaining against lamellocyte marker L1 in the circulating hemocytes (**A**–**D**) and lymph glands (**E**–**H**) of third-instar larvae. (**I**–**L**) The activity of the EGFR pathway was detected with anti-p-Erk antibodies. Immunostaining of circulating hemocytes (**I**,**J**) and lymph glands (**K**,**L**) showed high p-Erk (red) signals upon *gbb* knockdown. (**M**,**N**) Quantification of p-Erk intensity in circulating hemocytes (**M**) and lymph glands (**N**). In this and similar subsequent analyses, numbers of images of circulating hemocytes and numbers of lymph glands analyzed are indicated on histograms. (**O**–**R**) The activity of the JNK pathway was detected with anti-p-JNK antibodies. Immunostaining of circulating hemocytes (**O**,**P**) and lymph glands (**Q**,**R**) showed high p-JNK (red) signals upon *gbb* knockdown. (**S**,**T**) Quantification of p-JNK intensity in circulating hemocytes (**S**) and lymph glands (**T**). The arrows in B indicate the lamellocytes. Dashed white lines in (**I**,**J**,**O**,**P**) outline the edges of circulating hemocytes. For all quantifications: ** *p* < 0.01; *** *p* < 0.001; **** *p* < 0.0001 (Student’s *t* test). Scale bars: 50 μm (**A**–**D**), 100 μm (**E**–**H**,**K**,**L**,**Q**,**R**), 20 μm (**I**,**J**,**O**,**P**).

**Figure 3 cells-12-00661-f003:**
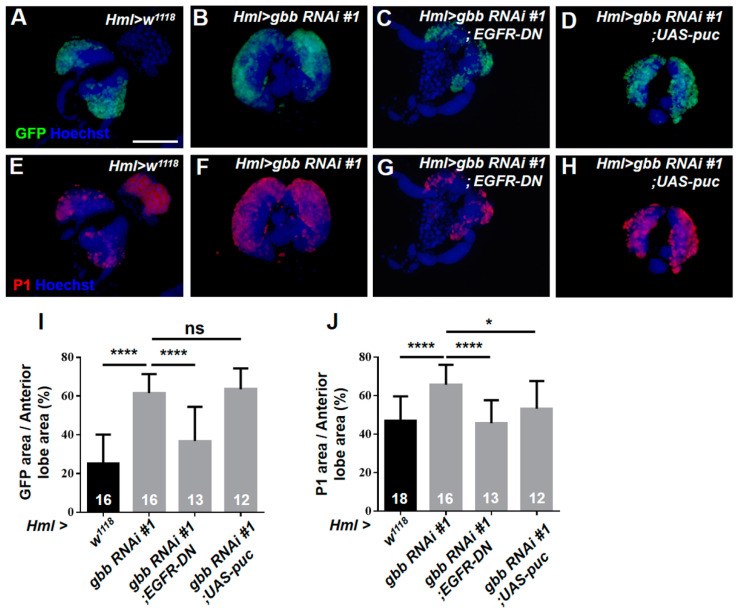
The knockdown of Gbb in the CZ induces plasmatocyte differentiation by activating the EGFR pathway but not the JNK pathway. (**A**–**D**) The Hml-GFP-positive area represents the area of the CZ. (**E**–**H**) Immunostaining for the plasmatocyte marker P1 showed that the P1-positive area was increased in the *Hml > gbb RNAi#1* lymph gland (**F**), and the aberrant plasmatocyte differentiation was rescued in *Hml > gbb RNAi #1 > EGFR-DN* (**G**) and *Hml > gbb RNAi #1 > UAS-puc* lymph glands (**H**). (**I**,**J**) Quantification of the proportions of the anterior lobes occupied by the Hml-GFP + area (**I**) and P1+ area (**J**), respectively. For all quantifications: ns, not significant; * *p* < 0.05; **** *p* < 0.0001 (Student’s *t* test). Scale bars: 100 μm (**A**–**H**).

**Figure 4 cells-12-00661-f004:**
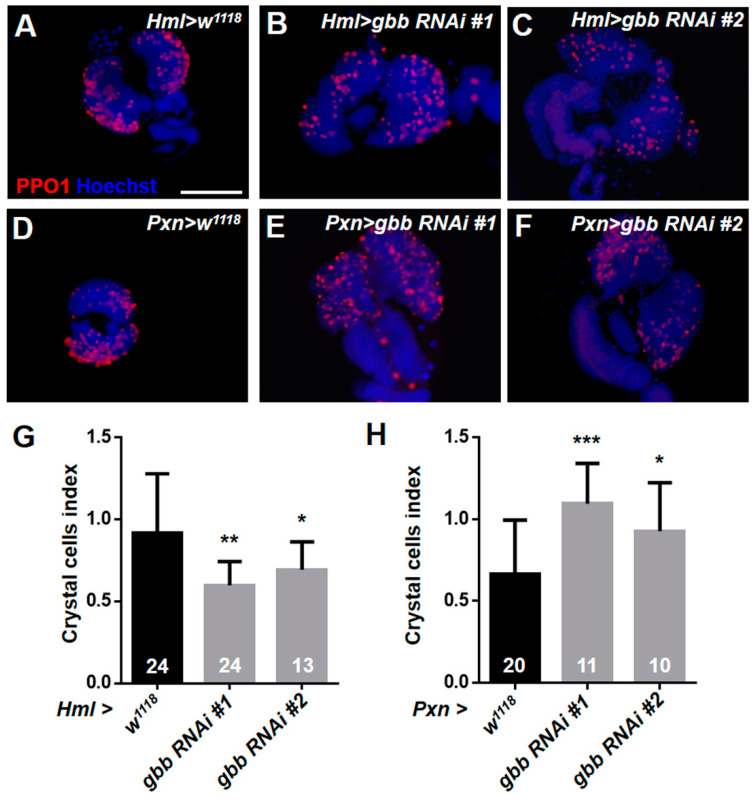
The knockdown of *gbb* in the CZ using the *Pxn-Gal4* driver causes the excessive differentiation of crystal cells. (**A**–**F**) Immunostaining of the lymph gland with anti-PPO1 antibody showed that using *Hml-Gal4* to knock down *gbb* reduced the differentiation of crystal cells (**A**–**C**), but using *Pxn-Gal4* to knock down *gbb* induced the excessive differentiation of crystal cells (**D**–**F**). (**G**,**H**) Quantification of the number of crystal cells. For all quantifications: * *p* < 0.05; ** *p* < 0.01; *** *p* < 0.001 (Student’s *t* test). Scale bars: 100 μm (**A**–**F**).

**Figure 5 cells-12-00661-f005:**
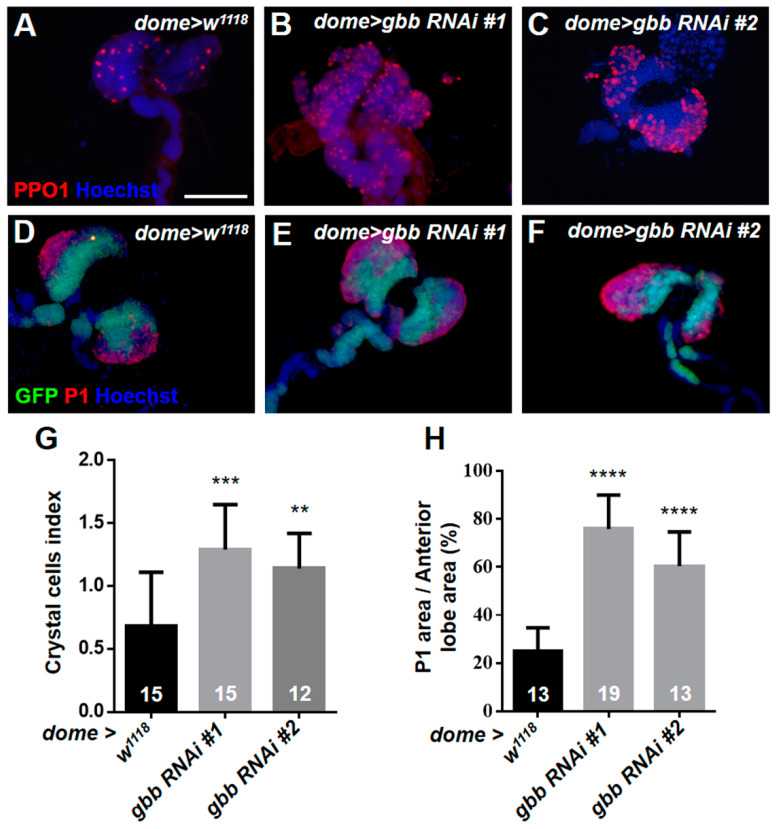
The knockdown of *gbb* in the MZ using the *dome-Gal4* driver results in the massive differentiation of crystal cells and plasmatocytes. (**A**–**C**) Immunostaining of lymph gland with anti-PPO1 antibody. (**D**–**F**) Immunostaining of lymph gland with anti-P1 antibody. The GFP signal area represents the area of dome > GFP+ MZ. (**G**,**H**) Quantification of crystal cell number (**G**) and P1-positive area (**H**). For all quantifications: ** *p* < 0.01; *** *p* < 0.001; **** *p* < 0.0001 (Student’s *t* test). Scale bars: 100 μm (**A**–**F**).

**Figure 6 cells-12-00661-f006:**
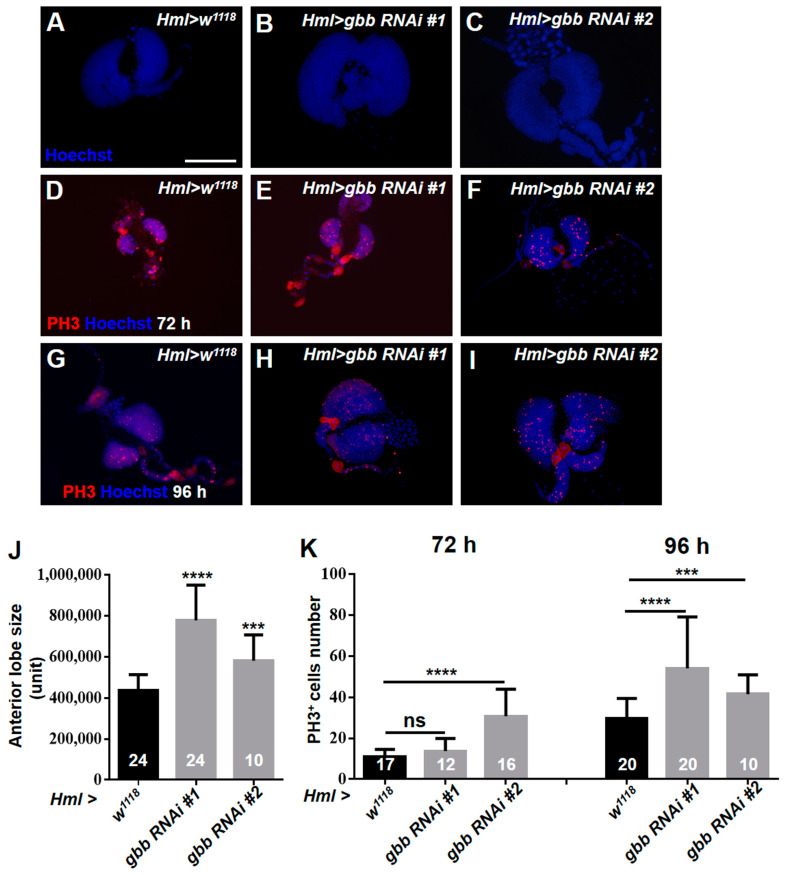
The loss of Gbb in the CZ induces the hyperproliferation of lymph glands. (**A**–**C**) Immunostaining of lymph glands with Hoechst showed that the anterior lobes of *Hml > gbb RNAi#1* and *Hml > gbb RNAi#2* lymph glands were obviously enlarged. (**D**–**I**) Immunostaining of lymph glands with anti-PH3 antibody showed increased PH3-positive cells in the *Hml > gbb RNAi#1* and *Hml > gbb RNAi#2* lymph glands at 72 h (**D**–**F**) and 96 h (**G**–**I**) after egg laying. (**J**) Quantification of anterior lobe size. (**K**) Quantification of PH3-positive cell number at 72 h (D–F) and 96 h (**G**–**I**) after egg laying. For all quantifications: NS, not significant; *** *p* < 0.001; **** *p* < 0.0001 (Student’s *t* test). Scale bars: 100 μm (**A**–**I**).

**Figure 7 cells-12-00661-f007:**
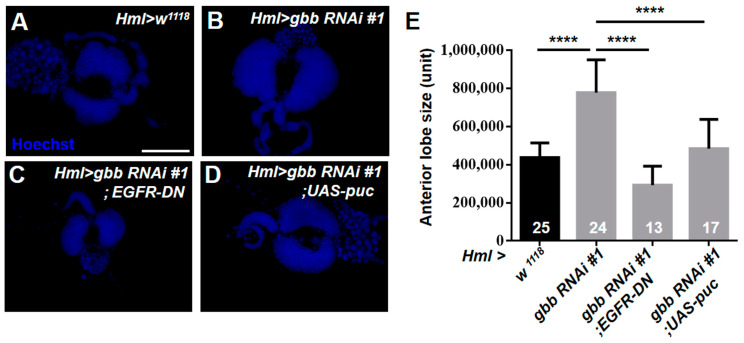
Inactivation of the EGFR or JNK pathway dramatically reduced the overgrowth of the anterior lobes in the *Hml > gbb RNAi#1* lymph glands. (**A**–**D**) Immunostaining of the lymph gland with Hoechst showed that the enlargement of the anterior lobes of *Hml > gbb RNAi#1* lymph glands was obviously rescued by the inactivation of EGFR and JNK pathways. (**E**) Quantification of anterior lobe size. For all quantifications: **** *p* < 0.0001 (Student’s *t* test). Scale bars: 100 μm (**A**–**D**).

## Data Availability

All data has been included in the manuscript.

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
