# Peer review of "Gbb Regulates Blood Cell Proliferation and Differentiation through JNK and EGFR Signaling Pathways in the Drosophila Lymph Gland"

_cells, 2023, doi:10.3390/cells12040661_

Round 1

Reviewer 1 Report

The manuscript by Zhang et al describes roles for the protein Glass bottom boat (Gbb), a BMP-like signaling molecule, in the lymph gland during Drosophila development.  The fly hematopoietic system is a good model system for studying blood cell development in general, and the high level of conservation between BMP signaling in flies and humans suggests that the pathway will have similar actions across organisms. Thus, the study may be of interest to a broad audience.

In this paper, the authors showed that a Gbb-GFP reporter is widely expressed in dissected lymph glands, consistent with prior RNA-seq data. Next, they disrupted Gbb function with tissue-specific expression of two RNA interference constructs, and found increased expression of a lamellocyte marker and higher levels of activated Erk and JNK.  Additionally, they see a wider expression pattern for a GFP reporter for the CZ in this genotype, and more plasmatocytes and crystal cells were identified.  They also show that the tissue undergoes more rounds of division and that the size of the tissue is generally larger when gbb is knocked down in it.  Several of the phenotypes were modified by blocking EGFR or JNK signaling, leading authors to conclude that Gbb works through these pathways to modify cell specification and proliferation in the lymph gland.

The paper is reasonably clearly written, and some of the data is convincing.  While some conclusions are well supported by the data, other conclusions are less substantiated and require more characterization of the existing data or additional experiments to support.  My concerns and suggestions are listed below.

Major concerns

Gbb-GFP in figure 1 looks nuclear in some cells, which I don’t think is expected.  What is the nature of this protein reporter, and how are the authors confident that the expression pattern is accurate?  It the expression affected by the RNAi construct?  If so, that may be a useful way to verify it.

The authors overexpress EGFR-DN or JNK along with Gbb-RNAi and find phenotypes differ from Gbb-RNAi alone. However, the appropriate controls are not shown, which precludes the ability to draw the main conclusions that the authors have.  The authors should show the EGFR-DN or JNK expressed in the tissue along with a control RNAi to see how those look alone (as strong or intermediate or weak) compared to the combination with Gbb-RNAi.  Conversely, they should show the Gbb-RNAi expression with co-expression of additional UAS constructs (like GFP) to show the RNAi expression is not reduced by the presence of multiple transgenes, which could also appear as a rescue.  Alternatively the authors could try different types of genetic interactions, such as combining hypomorphic alleles, to help support their conclusions.

From the data in figure 3, the authors state that “knockdown of Gbb in the CZ induces plasmatocyte differentiation by activating the EGFR but not JNK pathways”.  However, they see a statistically significant different (albeit small) in P1 area when both gbb is knocked down and JNK is inhibited, which argues against the conclusion.  Again, additional controls are also required to quantify this difference accurately.  So this conclusion is not well supported.

In figure 4 and 5, the authors show there are more cells with markers of crystal cells and plasmatocytes, however, they later show (figure 6) the lymph cells are undergoing more rounds of division.  Thus, it is not clear if the increase in certain cell types is simply due to there being more cells present  or if the proportion of certain cell types (due to changes in differentiation) is changing when gbb is disrupted.  These interpretations need to be clarified.

Minor concerns

There is a lot of jargon or abbreviations without explanation in the abstract and early text- (Dpp, CZ, EGFR, JNK, etc) these require definitions when they first appear.

Figure 1 appears to present data from a public, online database, but the way it is presented in the paper, it seems like the authors conducted this experiment themselves.  This needs to be made more evident and the key that is shown needs to be fully explained

Figure 2 – the circulating hemocytes are hard to see and could be outlined. It was also not very clear from the methods how they were obtained for staining or if Gbb-GFP is expressed in these cells. Also, wouldn’t p-ERK and p-JNK be expected to be nuclear?

Additionally, while the authors do show the stages of lymph gland development by time clearly in figure 6, it is less clear in the other figures, which could also impact size and number of cells specified.  If the time is not specified, are the tissues always from third instar? This should be made explicit.   

Reviewer 2 Report

The Authors presented exciting results regarding Gbb pathways inside of hematopoiesis of Drosophila, bringing understanding about BMP family protein regulating EGFR and JNK and consequent increase in cell growth. They used a strategy to get a response in the transgenic animals by analyzing specific proteins with GFP marked protein and silencing pathway proteins by RNAi knocking down a specific protein. They addressed the animal model with the right questions to follow the protein pathway, and so they found an increase of EGFR and JNK pathways could increase plasmatocyte number. The hematopoiesis animal model is a challenge in the scientific comunity because each organism has a different particularity when it considers blood formation. Getting other models could help to solve specie specific problems. Understanding how is Drosophila hematopoiesis and how its protein signaling works will make this model comparable and turn it into a valuable information source between Drosophila and mammals in the hematopoiesis field. Despite all the results being well presented, the methodology of measuring the proteins is a weakness because they may give variability in the intensity of fluorescence picked in the microscopy if that is not strictly controlled. Additional methodology such as Western blotting would be interesting to get more accurate conclusions and to make them stronger. But two points may difficult this analysis; one would be the concentration of protein enough for analysis, and the second, they could not track the marker location accurately. According to the methodology the authors described, it is accepted. Still, regarding the methods, they need to describe the RNAi protocol. If the authors describe the type of error shown (SD or SE) - SD would be preferred- plotted on the columns in the graphics and add, in the legend, the number of replicates used in the analysis, they could make their data more reliable. Indeed, the authors suggest that Gbb may be "involved in tumor development as a tumor repressor" which must be confirmed in the future into mammals.

Round 2

Reviewer 1 Report

Zhang et al describe how Glass bottom boat (Gbb), a BMP-like signaling molecule, acts in the lymph gland during Drosophila development.  Given the high level of conservation of BMP signaling, this study may inform research on blood cell development in general. Thus, the study may be of broad interest.  The authors have improved the clarity of the revised manuscript, explained methods in more detail, strengthened some of the analyses, and provided background information to reviewers that was helpful.  Thus, the study is now acceptable.

However, some minor revisions of the text are recommended: 

There are still a few places where abbreviations are not defined, eg, JAK-STAT.

The authors explain in the response to reviews why they believe the Gbb-GFP line is accurate, based on prior studies, but this should be explicitly explained, with references cited in the text.

In the response to reviews, the authors show that EGFR-DN or JNK expressed in the lymph gland doesn’t change the overall size, but they don’t show markers for differentiation.  I still think it would be helpful to include this result with differentiation markers along side of the combined Hml> RNAi and overexpression results in figure 3, so the different genotypes can be directly compared, but I understand these results may be known, so I will leave that decision to the authors. 

The figures have improved, but the legends need to be edited to describe the additions (eg, the meaning of the numbers on the bar graphs and what the arrows refer to).
